# Treatment of Infection-Related Non-Unions with Bioactive Glass—A Promising Approach or Just Another Method of Dead Space Management?

**DOI:** 10.3390/ma15051697

**Published:** 2022-02-24

**Authors:** Holger Freischmidt, Jonas Armbruster, Catharina Rothhaas, Nadine Titze, Thorsten Guehring, Dennis Nurjadi, Robert Sonntag, Gerhard Schmidmaier, Paul Alfred Grützner, Lars Helbig

**Affiliations:** 1Department of Trauma and Orthopedic Surgery, BG Trauma Center Ludwigshafen at Heidelberg University Hospital, 67071 Ludwigshafen am Rhein, Germany; holger.freischmidt@bgu-ludwigshafen.de (H.F.); jonas.armbruster@bgu-ludwigshafen.de (J.A.); catharina_97@gmx.de (C.R.); nadine.titze@gmx.de (N.T.); paul.gruetzner@bgu-ludwigshafen.de (P.A.G.); 2Trauma Centre, Hospital Paulinenhilfe Stuttgart at Tübingen University Hospital, Rosenbergstr. 38, 70176 Stuttgart, Germany; thorsten.guehring@diak-stuttgart.de; 3Department of Infectious Diseases, Medical Microbiology and Hygiene, Heidelberg University Hospital, Im Neuenheimer Feld 324, 69120 Heidelberg, Germany; dennis.nurjadi@med.uni-heidelberg.de; 4Laboratory of Biomechanics and Implant Research, Clinic for Orthopedics and Trauma Surgery, Heidelberg University Hospital, Schlierbacher Landstrasse 200a, 69118 Heidelberg, Germany; robert.sonntag@med.uni-heidelberg.de; 5Clinic for Orthopedics and Trauma Surgery, Center for Orthopedics, Heidelberg University Hospital, Schlierbacher Landstrasse 200a, 69118 Heidelberg, Germany; gerhard.schmidmaier@med.uni-heidelberg.de

**Keywords:** bioactive glass, animal model, non-union, bone defect, bone infection

## Abstract

The treatment of infected and non-infected non-unions remains a major challenge in trauma surgery. Due to the limited availability of autologous bone grafts and the need for local anti-infective treatment, bone substitutes have been the focus of tissue engineering for years. In this context, bioactive glasses are promising, especially regarding their anti-infective potential, which could reduce the need for local and systemic treatment with conventional antibiotics. The aim of this study was to investigate the osteoinductive and osteoconductive effects, as well as the anti-infectious potential, of S53P4 using a standardized non-union model, which had not been investigated previously. Using an already established sequential animal model in infected and non-infected rat femora, we were able to investigate bioactive glass S53P4 under realistic non-union conditions regarding its osteoinductive, osteoconductive and anti-infective potential with the use of µCT scans, biomechanical testing and histological, as well as microbiological, analysis. Although S53P4 did not lead to a stable union in the non-infected or the infected setting, µCT analysis revealed an osteoinductive effect of S53P4 under non-infected conditions, which was diminished under infected conditions. The osteoconductive effect of S53P4 remained almost negligible in histological analysis, even 8 weeks after treatment. Additionally, the expected anti-infective effect could not be demonstrated. Our data suggested that S53P4 should not be used in infected non-unions, especially in those with large bone defects.

## 1. Introduction

Treating non-unions has been an immense challenge for orthopedic surgeons for decades [1,2]. Although there has been plenty of research, and surgical interventions have gone through significant improvements, the physical suffering and psychological stress are still severe for patients [3,4]. Not to mention the socio-economical and financial implications they have on the health care system [5]. It is not surprising that non-union rates are widely dispersed in the literature since there are many risk factors, such as severity and location of the injury, age, nicotine abuse, diabetes or obesity [6]. Additionally, the type of therapy is of major interest for the outcome [7,8]. A femoral shaft fracture treated with an intramedullary nail has a 1.7% risk of developing a non-union [8], while a tibial fracture with a uniplanar external fixator has a non-union rate of 14% [9]. When it comes to open fractures and fracture infections, the healing process is even more complicated and often results in large bone defects [10]. The risk of infection after open fractures amounts to up to 50% [11], which is a major risk factor for non-healing. For example, in a Gustilo–Anderson Type III tibial shaft fracture, non-healing is found in 80% of cases [12].

Autologous bone grafts represent the current gold standard of non-union therapy [13]. It presupposes repeated surgical debridement, bone reconstruction and adequate stabilization, often with system antibiotics [14]. Autologous bone grafting requires an additional surgical intervention and it yields donor site morbidity [15]. Furthermore, it is limited in its quantity. Hence there is a need for bone graft substitutes with osteostimulative, osteoconductive and even perhaps with an antibiotic impact to have alternative, but equal, therapy options to autologous bone grafting [16]. Many approaches are being researched, including allograft or a demineralized bone matrix, cell-stimulating factors, as well as synthetic materials [7,17,18]. The main aim of a bone graft substitute is maintaining the space and stability of a scaffold while advancing bone healing [19].

Bioactive glasses, such as S53P4 (Bonalive Biomaterials Ltd., Turku, Finland), are bioactive materials that activate osteoblasts and, therefore, stimulate new bone formation by forming a silica gel layer with amorphous calcium phosphates on the glass surface, which crystalize to osteostimulative hydroxyapatite [20,21]. Furthermore, it potentially enhances angiogenesis [22,23,24]. In addition to serving as an osteoconductive scaffold, S53P4 is a promising possibility for treating bone defects [25,26,27]. Common approaches to gain an antibiotic effect are through loading carriers with antibiotics [28,29,30]. However, bioactive glasses show a different approach: the antibiotic effect is obtained through a rapid pH increase and a consequential osmotic effect [31,32,33]. This potentially qualifies them as a great alternative for treating osteomyelitis while avoiding drug resistance [26,34]. Several in vitro and in vivo studies exploring the spectrum of bioactive glasses have been conducted in the past [19,22,26,35]. However, the study populations and fractures were heterogenous and, thus, randomized, comparative, reliable and valid data were lacking.

Our group has previously established a sequential animal model for infection-related non-unions with segmental bone defects, which allowed the evaluation of therapeutic interventions in a genuine clinical simulation. [36]. To our knowledge, S53P4 has never been evaluated in a two-stage infectious non-union model. This study aimed to investigate the osteoinductive and osteoconductive effects, as well as the anti-infectious potential, of S53P4 using a standardized non-union model.

## 2. Materials and Methods

### 2.1. Preparation of Infecting Agent

*Staphylococcus aureus* subsp. *aureus* ATCC^®^ 49230™ has already been established in several animal models [36,37]. The agent was cultured in a tryptic soy broth (TSB) at 37 °C with 5% CO_2_ under constant shaking (200 rpm). At mid-log phase on the day of infection the bacterial pellet was harvested, washed with sterile phosphate buffered saline (PBS), resuspended in sterile PBS and adjusted to 0.5 McFarland standard (1.5 × 10^8^ CFU/mL). A concentration of 1 × 10^5^ CFU/mL bacterial suspension was achieved by dilution, from which 10 µL (=10^3^ CFU) was used for the in vivo infection.

### 2.2. Bioactive Glass (S53P4)

The synthetic bone graft Bonalive^®^ putty (Bonalive Biomaterials Ltd., Turku, Finland) is a bioactive glass S53P4 (53% SiO_2_, 23% Na_2_O, 20% CaO, 4%, P_2_O_5_ [mass percentage/wt %]), combined with polyethylene glycol and glycerine. It is provided as a paste, which can be used directly without any preparations.

### 2.3. Implants

The first osteosynthesis was achieved with stainless steel Kirschner wires (Synthes GmbH, Umkirch, Germany) with a diameter of 1.2–1.6 mm, depending on the diameter of the femur. An angle-stable polyacetyl plate osteosynthesis with six cortical screws (RISystem AG, Davos, Cologny, Switzerland) was used for the second surgery.

### 2.4. Groups

We have included sixty-five animals in four groups (Table 1).

### 2.5. Animals, Operative Procedure and Osteotomy Model

All experiments were approved by the Animal Experimentation Ethics Committee of the Regierungspräsidium Karlsruhe, Baden-Württemberg (35-9185.81/G-155/17). Sixty-five female, 3-month-old Sprague–Dawley rats (Charles River, Sulzfeld, Germany) were randomly divided into four groups, as described in Table 1.

The average weight was 285 g. The animals were held in type IV macrolon cages under standard laboratory conditions (12h-light-12h-dark cycles, room air temperature 22–24 °C, humidity 50–60%) in groups of 3–4 rats per cage. Access to water and food pellets was available ad libidum. There was a 2-week acclimatization period after arrival of the animals, before the first surgery took place.

A two-stage animal non-union model, previously established by our group [36], was used for this study. At first, an osteotomy was created and treated with K-wire osteosynthesis. Five weeks later, a second procedure was performed to switch to an angle-stable plate osteosynthesis to treat the developed non-unions. Surgery was performed under general anesthesia by weight-adapted subcutaneous injection of medetomidine (Dorbene Vet 1 mg/mL; Pfizer Deutschland GmbH, Berlin, Germany), midazolam (Midazolam HEXAL 5 mg/mL; al AG, Holzkirchen, Germany) and fentanyl (Fentadon Dechra 50 µg/mL; Dechra Veterinary Products Deutschland GmbH, Aulendorf, Germany). To prepare for surgery, the left hind leg was shaved and disinfected with alcohol. The animals were placed on drapes on a heating plate to hold their body temperature. The bodies were covered with sterile sheets. After blunt preparation throughout the skin and muscles, the femur was exposed and a 5 mm mid-diaphyseal full-thickness osteotomy was performed using a diamond disk (Dremel, Racine, WI, USA). Stainless steel K-wire with a diameter in the range 1.2–1.6 mm (Synthes, GmbH, Umkirch, Germany) was used to treat the defect with an intentionally rotational unstable osteosynthesis. Before applying the K-wire, 10 µL PBS was added to the medullary cavity in groups 1 and 3, and 10^3^ CFU *S. aureus* in 10 µL PBS in infected groups 2 and 4. Skin, fascia and muscles were sutured using an intracutaneous technique (Resolon^®^ 4/0 Ethicon, Norderstedt, Germany) and additionally adapted with skin clamps (Autoclip Clips, Heidelberg, Germany).

All animals received a second surgery five weeks after the first procedure. Surgical access was the same as previously described. After removing the K-wire a radical debridement was performed. Microbiological swabs and small tissue samples from the non-union were taken for further analysis. An angle-stable poly-acetyl plate (25 mm-long, 4 mm-wide, RISystem AG, Davos, Switzerland) with eight predrilled holes for angle-stable cortical screws was situated on the anterolateral surface of the femur. Three screws, proximal and distal to the osteotomy were each used to fix the plate through the predrilled holes, so a stable plate osteosynthesis with an approximately 5 mm defect was created. Soft tissue was dried and S53P4 was modulated to fill the gap. The suture was made analogical to the first procedure.

### 2.6. Follow-Up

Buprenorphine (0.3 mg/mL bw; Buprenovet^®^; Bayer AG, Leverkusen, Germany) was used as an analgesic medication perioperatively, as well as for the following four days every twelve hours. Body weight and body temperature, along with clinical conditions of the animals, were evaluated regularly. All animals were monitored for eight weeks after the second surgery and sacrificed.

### 2.7. μCT Scan Evaluation

μCT scans were performed with a Skyscan 1076 in vivo Micro-CT (Bruker, Kontich, Belgium), as previously described [38]. All animals had four scans in total. The first one was taken right before the second surgery to verify a non-union. After four and eight weeks, in vivo scans were performed for follow-up of the bone morphology. Immediately after sacrificing and removing the soft tissue, an ex vivo scan was performed to achieve a high-resolution image at the endpoint. The objects were scanned with a scan orbit of 360 degrees, an isotropic pixel size of 18 μm and energy settings of 100 kV (voltage), 280 ms (exposure time) and 100 μA (current) through a 1.0-mm aluminum filter. Rotation steps (1°/0.6°/0.4°) and frame averaging (2/4/6) were adjusted regarding the different types of scans (pre-op scan/4- and 8-week in vivo scan/ 8-week ex vivo scan). Image reconstruction was performed by using SkyScan NRecon software (v.1.6.9.8, Bruker microCT, Kontich, Belgium): ring artefact reduction (9/20), beam hardening correction (30%) and smoothing (1/10) were used as parameter settings. The contrast limits were set at 0–0.035 pre-op and for the ex vivo scan and 0–0.3 for the in vivo scans.

Two different scores by two independent observers each were assessed for qualitative evaluation of the datasets by simultaneously viewing a coronal, sagittal and transversal section in the SkyScan DataViewer (v.1.5.2.4, Brucker microCT, Kontich, Belgium).

The Lane and Sandhu scoring system was used to evaluate new bone formation, as previously described [39]. The modified An and Friedman Score was adapted to gauge the bone infection [37,40].

A SkyScan CTAnalyzer (v.1.13.21, Brucker microCT, Kontich, Belgium) was used to quantitatively evaluate the bone adjacent to the defect gap. To define the volume of interest (VOI), the center of the defect gap was determined visually and locations 3 mm proximal and distal from there were selected, which gave a total bone area of 6 mm with 351 images per scan. The transversal extension of the VOI, the region of interest (ROI), was specified by drawing semi-manually: a ROI was drawn manually every 10 to 15 images and the intervening images were interpolated, checked manually afterwards and adapted if necessary. Whilst the control group was left empty in the defect group (control group: VOI_control; intervention group: VOI_bone), the intervention group had two additional VOIs per scan: by interpolating between two ROIs from the bone ends, a second ROI was created, which contained S53P4 and represented the maximum width of the bone (VOI_all). After subtracting VOI_bone and VOI_all, the third VOI, including only S53P4 and soft tissue, was generated.

The following parameters were used for bone morphometry of the VOI for each in vivo scan (4 and 8 weeks after the second surgery): bone volume, bone surface, bone surface volume ratio, bone surface density, trabecular thickness and total porosity.

Density measurements were derived from the ex vivo scans. Two Phantoms with known density (0.25 and 0.75 g/cm^3^) were used to calibrate the gray values of the CTAnalyzer and measurement of bone mineral density (BMD) and tissue mineral density (TMD) was performed: TMD, which is equivalent to cortical bone, was measured by thresholding VOI_bone; BMD, which represents trabecular bone and bone marrow, was measured by analyzing the ROI of S53P4.

### 2.8. Microbiological Evaluation

As previously described, microbiological samples were taken with a 10mL wound eSwab™ (Becton Dickinson, Heidelberg, Germany) during the second surgery, as well as the sacrifice. A volume of 10 µL of the sample was inoculated onto Columbia agar supplemented with 5% sheep blood (Becton Dickinson, Heidelberg, Germany) and was cultured at 37 °C with 5% CO_2_ for 24 h on BD Columbia agar. Colonies consistent with *Staphylococcus aureus* morphology were confirmed as *S. aureus* by the slide agglutination test (Pastorex Staph Plus, Bio-rad, Germany) [41] and MALDI-TOF (Bruker GmbH, Germany). *Spa* typing (comparison of the polymorphic protein A gene) by Sanger sequencing was performed to confirm the strain identity with the infection strain [41].

### 2.9. Sacrifice

All animals were sacrificed eight weeks after the second surgery under general anaesthesia followed by CO_2_ in a sedation box. Following an in vivo μCT scan, the left femora were approached as described in the preceding surgeries. After a microbiological swab was acquired from the osteotomy, proceeding under unsterile conditions was possible. The left femur was disarticulated at both joints and the soft tissue was detached from the bone. For all animals determined to undergo biomechanical testing, the right femora were also dissected. All operated femora had an ex vivo μCT scan. Afterwards, the bones assigned for biomechanical evaluation were stored at −20 °C and the bones intended for histological evaluation were fixated in 4.5% paraformaldehyde (Roti-Histofix, Roth, Karlsruhe, Germany).

### 2.10. Mechanical Testing

The femora were biomechanically evaluated in a test device that measures the torsional stiffness of bones, as previously described [38]. The frozen femora were left for at least two hours at room temperature in saline solution to defrost. The plate osteosynthesis was removed. The distal and proximal parts of the bones were placed into two embedding molds (Technovit 4071, Heraeus Kulzer GmbH, Germany), while the defect region remained free for torsional testing. The lower embedding mold was connected to a pivotable axis, while rotation of the upper mold was restrained. The resulting maximum torque was recorded (8661-4500-V0200, Burster, Germany), while a linear constant rotation (20°/min) was applied by the testing device until a fracture occurred. In the case of a stable decrease, or reaching a maximum torque of 0.5 Nm, the recording was stopped. For comparison, all contralateral femora were tested as well.

### 2.11. Histology

After sacrifice, the femora were fixated in 4.5% paraformaldehyde (Roti-Histofix, Roth, Karlsruhe, Germany) for four days and were decalcified with ethylenediaminetetraacetic (Entkalker Soft, Roth, Karlsruhe, Germany) for 3 weeks. Plates and screws were removed and a graded alcohol series was made for bone dehydration (2 days, 70% ethanol; 2 days, 96% ethanol; 2 days, 100% ethanol). The bones were placed in acetone for 8 h to degrease and embedded in paraffin afterwards. The paraffin block was cut at 5 µm intervals at the longitudinal section of the bone down to the center of the sample. Sections were stained with haematoxylin and eosin (HE) (Carl Roth GmbH & Co KG, Karlsruhe, Germany) and brilliant-crocein-fuchsin acid, and safran (Pentachrom: Chroma-Waldeck GmbH & Co. KG, Münster, Germany) for overview staining. Furthermore tartrat-resistent acid phosphatase (TRAP) staining (Merck KGaA, Darmstadt, Germany) and Toluidine blue staining (Sigma-Aldrich Chemie GmbH, Steinheim, Germany) were performed. In addition, immunohistochemical staining was performed with anti-CD14 (ab203294, Abcam, Cambridge, UK), anti-CD31 (ab182981, Abcam, Cambridge, UK) and anti-CD68 antibodies (ab125212, Abcam, Cambridge, UK).

All samples were provided by the Tissue Bank of the National Center for Tumor Diseases (NCT) Heidelberg, Germany, in accordance with the regulations of the tissue bank and the approval of the ethics committee of Heidelberg University. Fiji ImageJ (v.1.53c) was used for analysis. A 6mm ROI, equivalent to the CT-VOI, was selected. The background was subtracted to minimize artefacts and a specific color threshold for each staining was applied (Table 2). The selected area, as well as the total bone area (hue 0/255, saturation 0/255, brightness 0/250), was measured.

### 2.12. Statistical Analysis

Data were recorded in Excel (Microsoft, Redmond, WA, USA) and statistically analyzed via GraphPad Prism version 9.1.0 (GraphPad Software, San Diego, CA, USA). The D’Agosino–Pearson test was employed to evaluate for normal distribution. To test for the statistical significance of differences between the four groups, one-way analysis of variance (ANOVA) followed by Tukey’s multiple comparisons test were performed. The µCT data at the two time points (four weeks and eight weeks postoperative) were compared using the Student’s paired *t*-test. *p*-values < 0.05 were considered statistically significant. All tests were performed two-sided. Data were presented as mean ± standard deviation (SD) in the figures and throughout the manuscript unless otherwise indicated.

## 3. Results

### 3.1. Failure Parameters

Two rats died in the NI control group during anesthesia during the first intervention. Four rats of the NI S53P4 and one rat of the I S53P4 group died during the second intervention. One animal in the I S53P4 group had to be euthanized due to unconscionable wound conditions. Fifty-seven of the sixty-five animals reached the endpoint of the study.

Six animals from the NI control group and five animals from the NI S53P4 group had to be excluded due to a secondary bone infection. As previously described, a secondary bone infection was determined by a positive microbiological result and an An and Friedman score ≥19 [36]. Positive microbiological results with an An and Friedman Score <19 were considered as colonization.

Additionally, 3 CT scans in the 4-week I control group and 1 CT scan in the 8-week I control group were excluded due to technical errors during the imaging process.

### 3.2. Microbiologic Results Showed No Anti-Infective Potential in S53P4

Microbiologic analysis at two time points (second surgery and euthanasia) showed more sterile results in the non-infected groups but some colonization and secondary infection with *S. aureus*. All bacteria detected in the non-infected groups had a different *spa*-typing than the inoculated strain. Infected groups showed the expected infection with the inoculated *S. aureus strain*. No clear pattern, leading to more sterile results in the S53P4 groups, could be detected. Furthermore, the infected S53P4 group showed even more variety in bacterial load with the detection of Enterobacterales (Enba) (Figure 1).

### 3.3. S53P4 Did Not Lead to a Stable Union in Non-Infected or Infected Conditions

Biomechanical testing of the femora was performed by measuring the maximum torque. In all the measured groups, the maximum torque was significantly lower than in the contralateral femora (CF NI control, 0.45 Nm) (Figure 2). No significant difference was detected in the ipsilateral femora between the control and S53P4 groups, regardless of the treatment in non-infected or infected conditions (Figure 2); thus, S53P4 alone did not lead to functionally relevant stability of the bone.

### 3.4. µCT Analysis of the Bone Showed an Osteoinductive Effect of S53P4 in Non-Infected Groups, Which Was Diminished in Infected Groups

µCT analysis of the bone 4 and 8 weeks after the second surgery revealed an osteoinductive effect of S53P4 in non-infected conditions (Figure 3A) with a significant increase in bone volume and a decrease in porosity. This effect was already detectable at the 4-week timepoint but was even more apparent at the 8-week timepoint. Under infected conditions, S53P4 also showed an osteoinductive effect, which could not reach the potential in the non-infected groups (Figure 3A).

The density of the bone and the bone graft S53P4 was measured via density analysis during the CT scans in the segmented regions corresponding to bone tissue following the tissue mineral density (TMD) protocol of the analysis software. The bone density was significantly lower in infected groups than in the non-infected counterparts. No significant difference was seen between the control and S53P4 groups (Figure 4a). S53P4 density measurements (bone mineral density—BMD) showed a significant decrease in the infected samples (Figure 4b)**.**

### 3.5. µCT Analysis of S53P4 Showed the Detrimental Effects of Infection

µCT analysis of S53P4 4 and 8 weeks after the second surgery showed a significant decrease in the bone graft volume just 4 weeks after the surgery, which further decreased significantly until 8 weeks postoperation. In contrast, the bone graft volume in the non-infected group even increased between 4 and 8 weeks leading to an even further gap between those two groups (Figure 5). Correspondingly, the surface volume ratio increased in I S53P4 while the surface density decreased. Lastly, the porosity rose to nearly 100%, indicating that the S53P4 was almost gone at that point (Figure 5).

### 3.6. There Was a Lower Invasion of Osteoclasts into the Bone Defect in Infected Groups but Also in Non-Infected S53P4

TRAP Staining of the whole femora showed an invasion of osteoclasts at the border between the bone ends and the defect in the NI control. This effect could not be detected in the other groups, indicating an inhibiting effect of either infection and/or S53P4 (Figure 6).

### 3.7. Infection-Triggered Hypervascularization Was Diminished in I S53P4

Immunohistochemical analysis comparing the vascularization was performed using the marker CD31. The I control showed significant hypervascularization compared to the NI control. This increase could also be seen in S53P4 but infection-triggered hypervascularization was diminished by roughly 50% (Figure 7).

### 3.8. No Reduction in the Inflammatory Response in S53P4 in CD14 and CD68 Staining

To analyze the systemic inflammatory response, CD14 and CD68 stainings were prepared. Both CD14 and CD68 are markers for macrophages and correlate with the inflammatory response and the anti-infective potential of the bone graft substitute. In concordance with the results already described in 3.5, no reduction in inflammatory response was found in S53P4 (Figure 8 and Figure 9).

## 4. Discussion

The presented study examined the role of the bioactive glass S53P4 in a sequential non-union animal model [36] under non-infected and infected conditions. Due to the similarities between the two-step model in rats to the approach used to treat non-unions in patients, we were able to test S53P4 in a standardized environment with high translatability into day-to-day clinical work.

We aimed to test the potential of S53P4 for eradicating bacterial infections or keeping non-infected environments aseptic. Furthermore, its potential to induce new bone and, thus, biomechanical stability was examined. In this context osteoconduction and osteoinduction, which are part of the widely recognized diamond concept for healing long bone non-unions, are major aspects addressed in our study [14].

Regarding the antiseptic effect of S53P4—although there are widely acknowledged antimicrobial effects of S53P4 in vitro [42,43]—we could not confirm eradication of *S. aureus* in any of the animals at euthanasia in our microbiologic testing. In addition, bacterial detection increased after the implantation timepoint in both NI groups and it strongly increased in NI S53P4. Thus, our data questioned the role of S53P4 in the treatment of bone infections as well as in the prevention of secondary infections in prior sterile conditions. Of note, is the fact that our microbiological results are based on intraoperative swabs while the gold standard would be the microbiological processing of tissue samples. This could lead to some false negative samples due to lower sensitivity. As we used this method in all our groups, errors between the different conditions could be ruled out. In line with this, staining for markers on immune cells, such as CD14 and CD68, were increased in the infected groups compared to their non-infected counterparts, indicating ongoing osteitis and systemic responses of the immune system. There was no significant reduction with the addition of S53P4, underlining that infection could indeed not be eradicated. This was in line with the results of other groups who also described insufficient eradication by S53P4 [44]. Due to the structure of S53P4, the histological preparation may be impaired due to difficult slicing.

Another important observation was the detrimental effects of the infection on the bone graft S53P4 with a significant reduction in size and a dramatic increase in porosity 4 weeks after surgery, with a significant worsening 8 weeks after surgery, indicating ongoing degradation of S53P4 due to bacterial overload (Figure 5). This effect could not be seen under aseptic conditions, making it unlikely to be due to beneficial cell invasion.

At least in this setting of highly active osteitis, S53P4 could not meet the expectations generated by reports in vitro [31] and in patients [26,45]. One explanation for this discrepancy could be the missing systemic antibiotic therapy in our model, which would have been administered as part of the standard patient treatment [45]. Non-compliancy of the investigated animals could also have affected our results with a mentionable incidence of wound healing disorders. Nevertheless, these results also question the use of S53P4 for implant coatings on metallic implants [46] to prevent secondary infections.

In biomechanical testing, neither the absence nor the presence of S53P4 led to stable unions under non-infected or infected conditions. We would like to mention that stable unions were not part of the expected results as our study primarily focused on analyzing the healing process itself rather than confirming union at the endpoint. 

Interestingly µCT scans showed positive effects of S53P4 on the surrounding bone tissue with increased bone volume, indicating an osteoinductive effect of the bone graft substitute, at least under non-infected conditions as seen by other groups [47,48]. This early osteoinductive effect of S53P4 could even be detected in infected conditions 4 weeks after the second surgery. It is important to note that there was no further increase in bone volume under infected conditions between 4 and 8 weeks post second surgery, which matched the results of the examination of the bone graft substitute itself where infection led to the degradation of S53P4. Further analysis revealed a decrease in porosity in the bone tissue, which could indicate the induction of sclerotic rather than vital bone tissue. Regarding the osteoconductive potential of S53P4, one could argue that the increased porosity of S53P4 could also appear due to cell invasion. However, TRAP staining showed that although osteoclastic invasion was triggered in the tNI control due to he endogenic repair mechanisms, this activation was diminished under infected conditions but also due to the addition of S53P4. Our results conflicted with the findings in patients, where S53P4 was found to be as good as autologous bone tissue regarding osteointegration [25]. A possible explanation could be the inhibition of osteoclasts by silica, a component of S53P4, which was shown by Van Gestel et al. [49]. Another important difference could be the segmental bone defect used in this study in contrast to the cavitary defects S53P4 that were used in patients, as we know that cavitary defects have different local conditions that potentially facilitate the pH-sensitive antimicrobial effect of S53P4 [19].

Sufficient vascularization is one of the major requirements for an effective immune response and the bone healing process [50,51,52]. Endothelial marker CD31 was increased under infected conditions, indicating infectious hyperemia. Surprisingly, this effect was also seen in I S53P4 animals, but the effect was almost cut in half, indicating a possible inhibition of vascularization due to S53P4. Together with the histological results, this aggravated the impression that S53P4 acted as an inorganic barrier. This trend should lead to further exanimation of vascularization in the use of S53P4 due to its crucial necessity for unions and favorable patient outcomes. Further research is necessary to examine this possible threat.

In the future, the combination of favorable approaches could be of great interest. The combination of the osteoinductive effect of S53P4 and systemic agents, which enhance vascularization and/or osteoconductivity, such as PTH [53] or bisphosphonates, could lead to better outcomes. Another promising approach for the treatment of large bone defects could be the combination of S53P4 with autologous/allogenic bone grafts and local or systemic antibiotics. Newly designed bioactive glasses containing high concentrations of lithium ions [54] or copper-doped glasses [55] could also be interesting materials to study in this context. Their altered properties compared to S53P4 could lead to more osteoconductive or osteoinductive potential. Further research is necessary to address those promising bone graft candidates. Longer observation periods could also show differences in biomechanical stability, which our 8-week period could not show.

## 5. Conclusions

For the first time, S53P4, a bioactive glass already in clinical use, could be investigated in a realistic non-union environment. S53P4 has indicated osteoinductive potential in the treatment of aseptic non-unions. Osteoconduction or biomechanical stability could not be verified on segmental bone defects. Only a minor osteoinduction and osteoconduction was observed in septic non-unions, while the bone substitute was degraded over time. Our results indicated that S53P4, as a solitary treatment option, was not a promising approach for (infected) non-unions.

## Figures and Tables

**Figure 1 materials-15-01697-f001:**
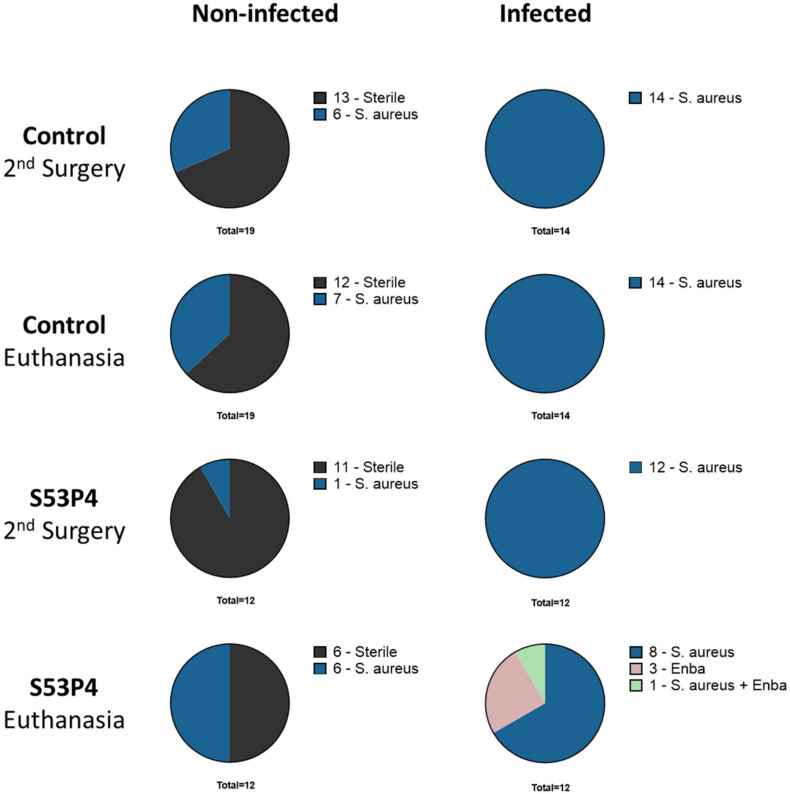
Microbiologic results of all animals after second surgery and euthanasia. Rats with secondary infection and colonization are both displayed as part of the *S. aureus* group. Abbreviations: Enba = Enterobacterales.

**Figure 2 materials-15-01697-f002:**
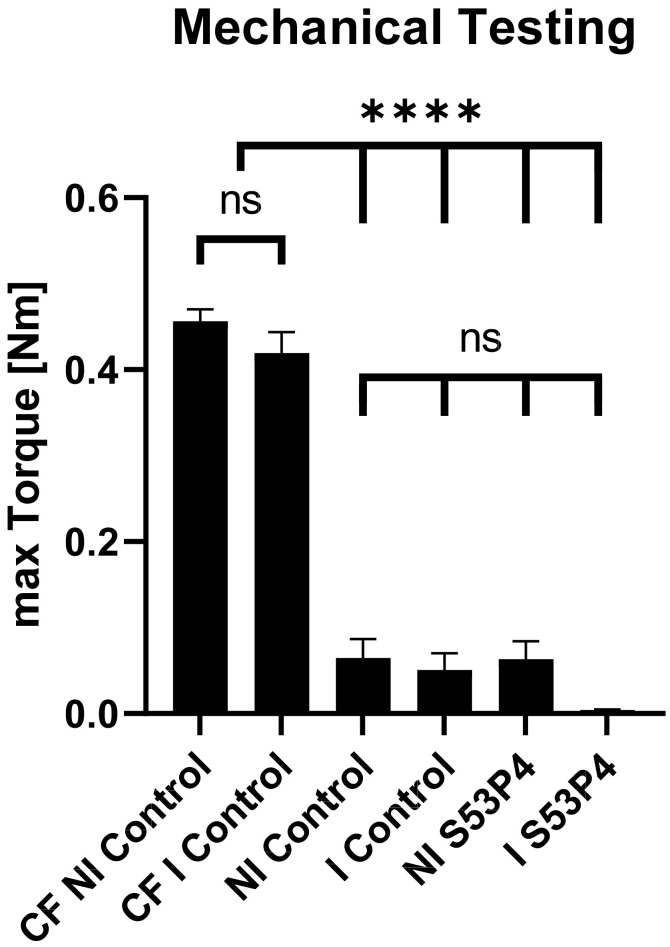
Mechanical testing. Max. torque of the non-infected contralateral femora (CF NI control) was significantly higher than the max. torque of all other groups. No significant differences between non-infected (NI) and infected (I) groups or the control and S53P4 groups were detected. **** = *p* < 0.0001; ns = not significant.

**Figure 3 materials-15-01697-f003:**
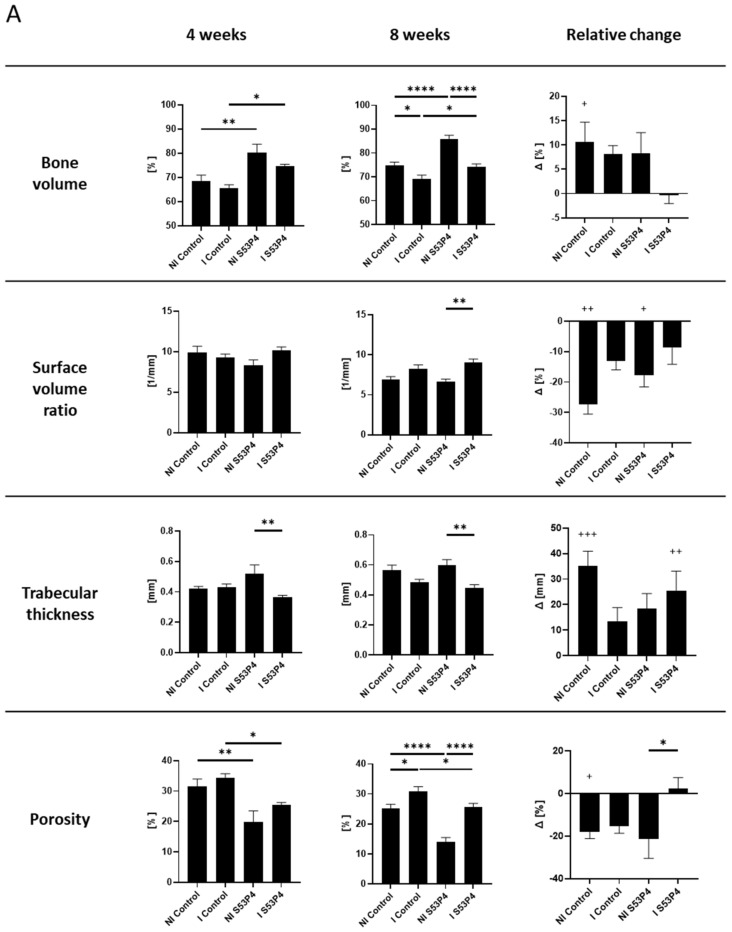
µCT analysis of the bone 4 and 8 weeks after the second surgery. (**A**) S53P4 led to an increase in bone volume after 4 and 8 weeks in non-infected and infected conditions but performed significantly worse in the infected groups in comparison to the non-infected groups. (**B**) µCT image coronary slide 8 weeks after the second surgery. **** = *p* < 0.0001; ** = *p* < 0.01; * = *p* < 0.05; +++ = *p* < 0.001, comparison of 4 to 8 weeks; ++ = *p* < 0.01, comparison of 4 to 8 weeks; + = *p* < 0.05, comparison of 4 to 8 weeks.

**Figure 4 materials-15-01697-f004:**
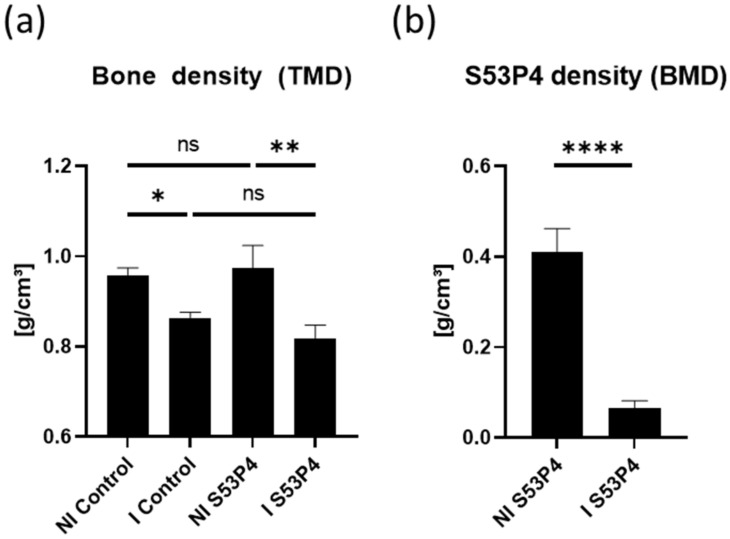
Analysis of bone density (TMD) and the density of S53P4 (BMD). (**a**) Infected groups showed less density in the control and S53P4 groups. (**b**) S53P4 density decreased significantly in the infected samples. **** = *p* < 0.0001; ** = *p* < 0.01; * = *p* < 0.05; ns = not significant.

**Figure 5 materials-15-01697-f005:**
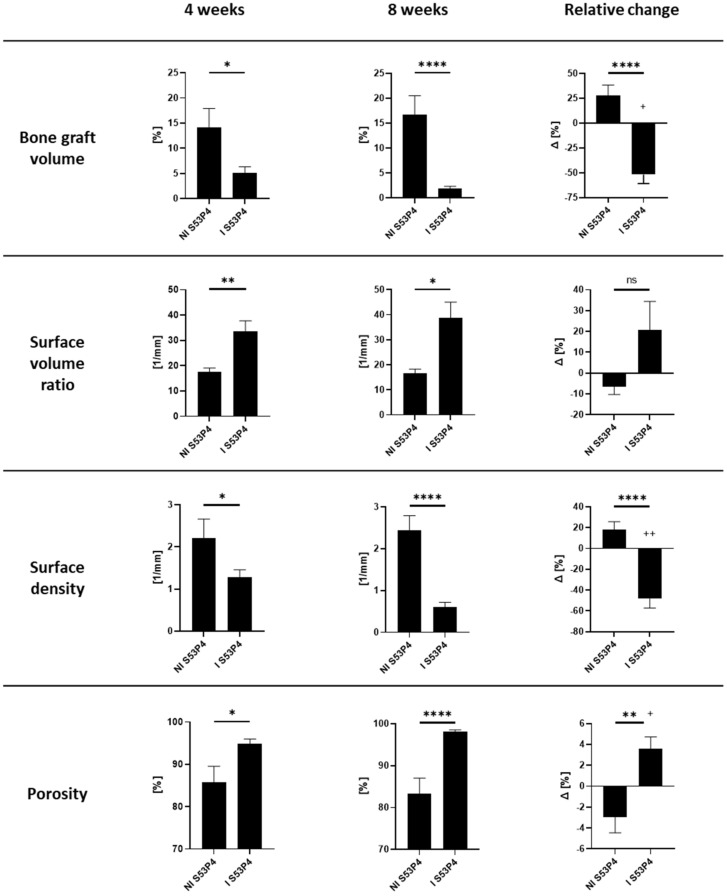
µCT analysis of the bone graft S53P4 4 and 8 weeks after the second surgery. The S53P4 was almost gone in the infected samples, especially 8 weeks after surgery. **** = *p* < 0.0001; ** = *p* < 0.01; * = *p* < 0.05; ++ = *p* < 0.01; + = *p* < 0.05; ns = not significant.

**Figure 6 materials-15-01697-f006:**
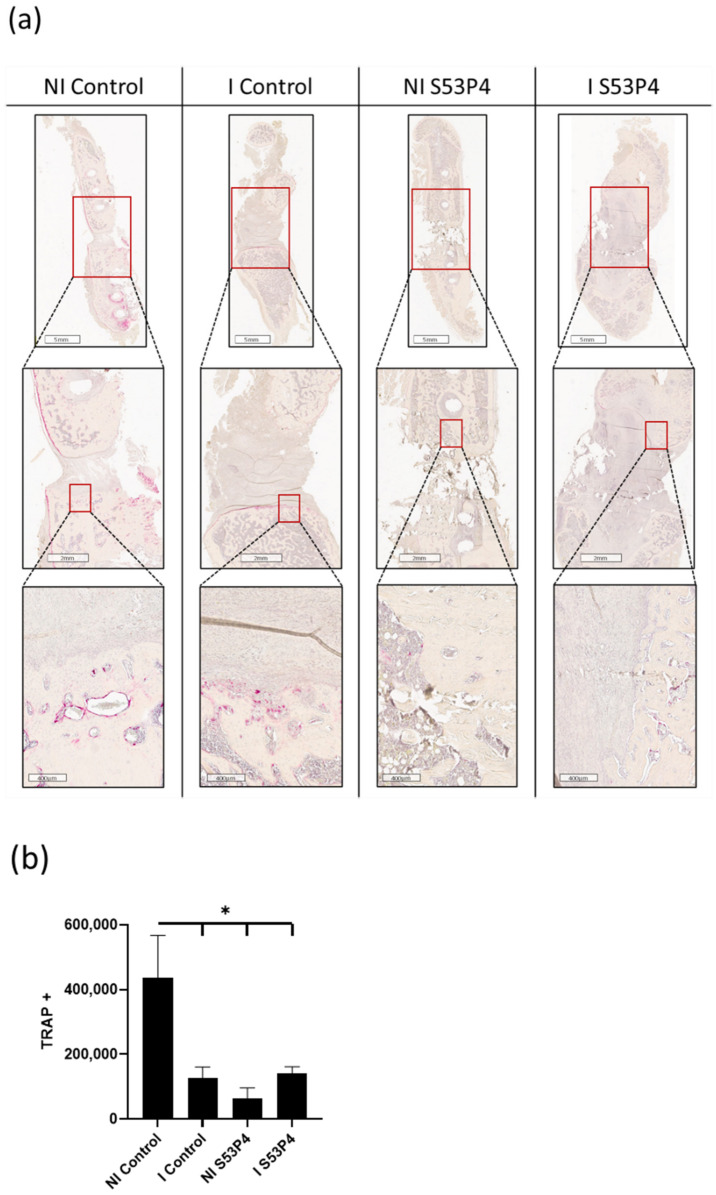
TRAP staining shows invasion of TRAP-positive osteoclasts in the defect in the NI control group but significantly less in the other groups. (**a**) TRAP staining (osteoclasts with red staining). (**b**) Quantification. * = *p* < 0.05.

**Figure 7 materials-15-01697-f007:**
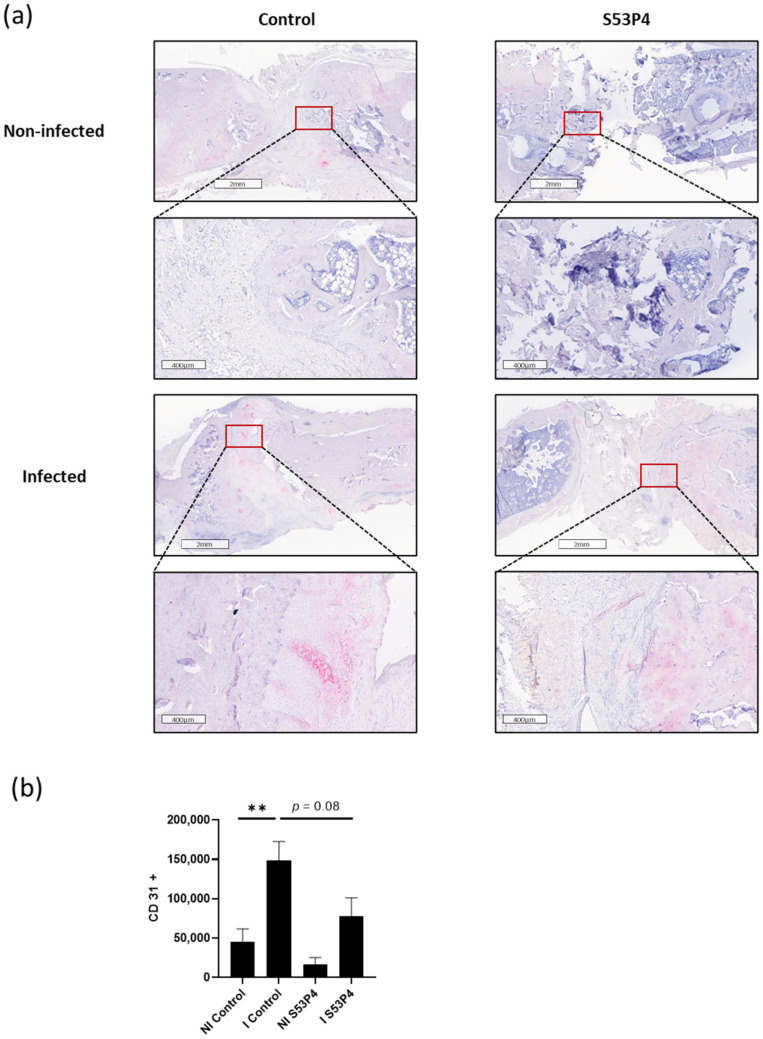
CD31 staining, a marker for vascularization. (**a**) Comparison of representative parts of the bone defect with magnification of the bone—defect border. Infection led to significant hypervascularization in the control groups. (**b**) Quantification. ** = *p* < 0.01.

**Figure 8 materials-15-01697-f008:**
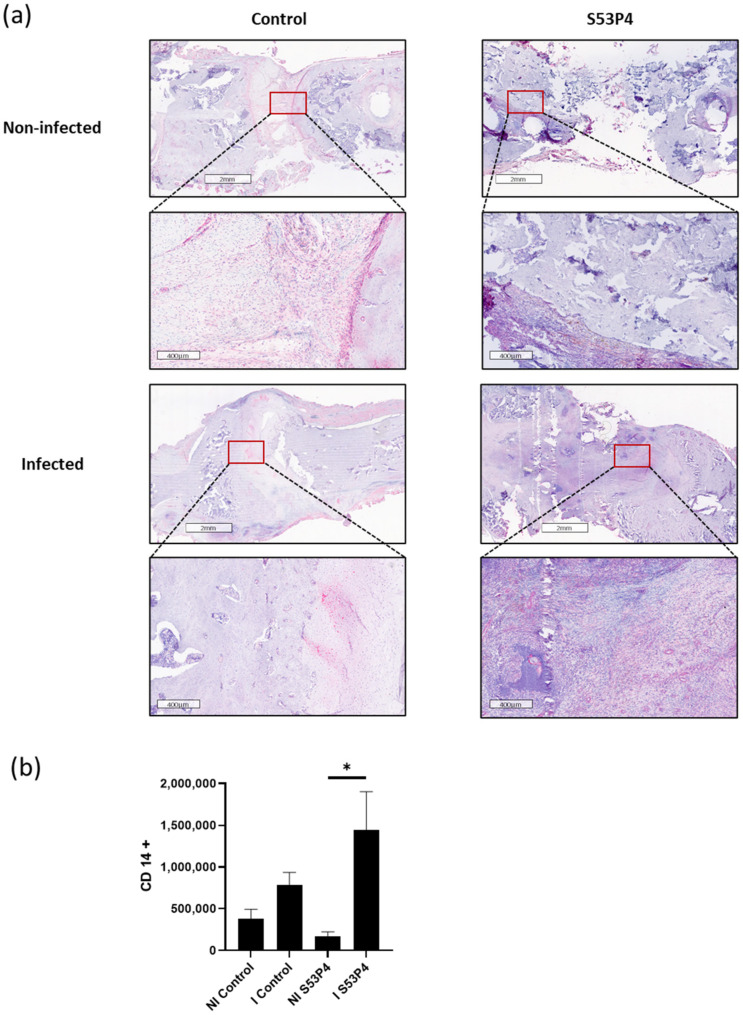
CD14 staining, a marker for macrophages. (**a**) Comparison of representative parts of the bone defect with magnification of the bone—defect border. There was a significantly increased infiltration of macrophages in I S53P4. (**b**) Quantification. * = *p* < 0.05.

**Figure 9 materials-15-01697-f009:**
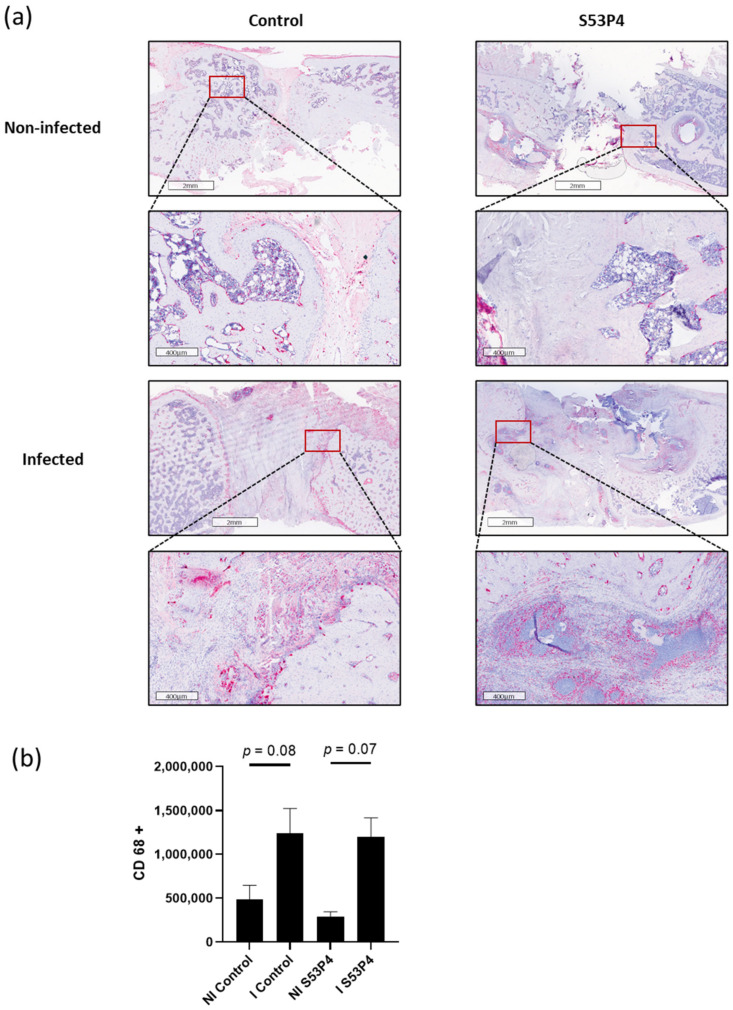
CD68 staining, a marker for macrophages. (**a**) Comparison of representative parts of the bone defect with magnification of the bone—defect border. There was an increased infiltration of macrophages in I control and I S53P4 without significance. (**b**) Quantification.

**Table 1 materials-15-01697-t001:** The following groups were examined.

Groups	Procedure
1: NI control group (non-infected group, n = 21)	K-wire osteosynthesis with intramedullary application of 10 µL PBS (non-infected)/10^3^ KBE *S. aureus* in 10 µL PBS (infected). Debridement and re-osteosynthesis after 5 weeks with an angle-stable plate.
2: I control group (infected control group, n = 14)	
3: NI S53P4 (intervention group, non infected, n = 16)	K-wire osteosynthesis with intramedullary application of 10 µL PBS (non-infected)/10^3^ KBE *S. aureus* in 10 µL PBS (infected). Debridement and Re-osteosynthesis after 5 weeks with an angle-stable plate and application of S53P4 to the bone defect.
4: I S53P4 (intervention group, infected, n = 14)	

**Table 2 materials-15-01697-t002:** Specific color threshold for each staining.

	Hue	Saturation	Brightness
CD31	200/255	40/255	0/255
CD14	200/255	60/255	0/255
CD68	200/255	100/255	0/255
TRAP	200/255	90/255	0/255

## Data Availability

The datasets used and analyzed during the current study are available from the corresponding author on reasonable request.

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
