# Peer review of "Treatment of Infection-Related Non-Unions with Bioactive Glass—A Promising Approach or Just Another Method of Dead Space Management?"

_materials, 2022, doi:10.3390/ma15051697_

Round 1
Reviewer 1 Report
The presented research is impressive in terms of design, implementation, identification and avoidance of potential sources of error.
It is noteworthy that bone repair is dependent on mechanical stimulation, along with infectious factors. Moreover, the local mechanical stimulation depends on the muscular force and the application surface, which can lead to the modification of the osteolysis at the level of the K-wires, if several diameters are used. It would be interesting to analyze the stratified data according to this diameter, but it is possible that the small number of laboratory animals used does not lead to statistical significance. It is possible that choosing the diameter of the K-wire as fixed element to lead to slightly different results.
Reviewer 2 Report
Please see the attached File

Reviewer 3 Report
Dear authors,
Thank you for submitting an original article focused on bioactive glass (S53P4), concerning its osteoinductive, -conductive and anti-24 infective potential. Presented below are my comments on the present manuscript:
- The percentages for bioactive glass S53P4, outlined in the line 89, are indicating wt% or some other ratio?
- The language throughout the manuscript is quite good, however there are some instances where it requires additional correcting (e.g. line 268). I would advise the authors to read through the text in entirety one more time. Additionally, for formal writing it is suggested to use the full negative form of the verb and not the abbreviated form as it can be found within the manuscript (e.g. didn’t should be written as did not, line 277 and 282).
- For results regarding mechanical testing, it would be beneficial to mention what is the generally accepted value of Max. torque that would indicate a favorable result (section 3.2).
- Considering that within the manuscript it was observed and stated that the infected group with S53P4 also showed an osteoinductive effect, but not as big as the non-infected groups (line 295), in the conclusion should be stated the same, and not that it was not observed at all (line 443).
